ecology, behaviour

competition, behavioural plasticity, intra-guild predation, circadian rhythm, elasmobranch, accelerometer

**Author for correspondence:**
Karissa O. Lear
e-mail: k.lear@murdoch.edu.au

# Temporal niche partitioning as a novel mechanism promoting co-existence of sympatric predators in marine systems

Karissa O. Lear[1], Nicholas M. Whitney[3], John J. Morris[4] and Adrian C. Gleiss[1,2]

[1]Centre for Sustainable Aquatic Ecosystems, Harry Butler Institute and [2]Environmental and Conservation Sciences, Murdoch University, 90 South Street, Murdoch, Western Australia 6150, Australia
[3]Anderson Cabot Center for Ocean Life, New England Aquarium, 1 Central Wharf, Boston, MA 02110, USA
[4]Mote Marine Laboratory, 1600 Ken Thompson Parkway, Sarasota, FL 34236, USA

 KOL, 0000-0002-2648-8564; NMW, 0000-0001-8797-6927; ACG, 0000-0002-9960-2858

Niche partitioning of time, space or resources is considered the key to allowing the coexistence of competitor species, and particularly guilds of predators. However, the extent to which these processes occur in marine systems is poorly understood due to the difficulty in studying fine-scale movements and activity patterns in mobile underwater species. Here, we used acceleration data-loggers to investigate temporal partitioning in a guild of marine predators. Six species of co-occurring large coastal sharks demonstrated distinct diel patterns of activity, providing evidence of strong temporal partitioning of foraging times. This is the first instance of diel temporal niche partitioning described in a marine predator guild, and is probably driven by a combination of physiological constraints in diel timing of activity (e.g. sensory adaptations) and interference competition (hierarchical predation within the guild), which may force less dominant predators to suboptimal foraging times to avoid agonistic interactions. Temporal partitioning is often thought to be rare compared to other partitioning mechanisms, but the occurrence of temporal partitioning here and similar characteristics in many other marine ecosystems (multiple predators simultaneously present in the same space with dietary overlap) introduces the question of whether this is a common mechanism of resource division in marine systems.

## 1. Introduction

Niche partitioning is one of the main mechanisms allowing sympatric competitors to coexist through the division of resources. Niche partitioning commonly takes several forms, including resource partitioning, where species specialize in different food or prey items, spatial partitioning, where species use different areas to forage or hunt, and temporal partitioning, where sympatric species rotate peak foraging times on a diel or seasonal scale [1]. Some partitioning mechanisms are promoted by morphological specializations of the species involved. For example, the evolution of different dentition and jaw morphologies in carnivore guilds promotes specialization of different prey resources [2,3], and adaptations of co-occurring species that restrict diurnal or nocturnal activity promote temporal partitioning (e.g. [4,5]). Additionally, within morphological or physiological constraints, partitioning regimes can also shift behaviourally based on environmental variation or changes in assemblages of competitor guilds or predator and prey species [6–9].

It is particularly important to understand the partitioning mechanisms that drive coexistence of predator populations; predators generally disproportionately affect ecosystems through top-down control and behaviourally mediated impacts on prey species [10,11], and the maintenance of predator populations is necessary for the preservation of healthy ecosystems [12]. Given the high rates of anthropogenically driven environmental change, it is also crucial to understand the current patterns and drivers of partitioning in predator guilds; shifts

**Table 1.** Tagging metadata for each shark species. The number and sizes of individuals of each species used in analyses, along with the total data volume used, the range of water temperatures recorded, the range of hourly mean overall dynamic body acceleration (ODBA) and the approximate timing of peak activity (greater than 80% of the difference between their minimum and maximum ODBA) identified by generalized additive mixed models. Note that only data from winter months when all species concurrently inhabit the study area were used (approx. 16–26°C, see data temperature range below), but most species were also present in the study area at warmer temperatures.

| species | N | hours of data | TL range (cm) | data temperature range used (°C) | ODBA range (g) | timing of peak activity |
|---|---|---|---|---|---|---|
| blacktip shark *Carcharhinus limbatus* | 21 | 500 | 126–186 | 18.8–27.2 | 0.02–0.19 | 18.00–21.00 |
| bull shark *Carcharhinus leucas* | 11 | 260 | 181–269 | 19.8–26.0 | 0.02–0.11 | 4.00–10.00 |
| sandbar shark *Carcharhinus plumbeus* | 71 | 1676 | 162–227 | 16.2–26.5 | 0.02–0.08 | 13.00–19.00 |
| tiger shark *Galeocerdo cuvier* | 39 | 827 | 154–264 | 16.2–23.5 | 0.02–0.10 | 9.00–15.00 |
| great hammerhead *Sphyrna mokarran* | 15 | 264 | 205–292 | 23.5–26.3 | 0.06–0.17 | 21.00–03.00 |
| scalloped hammerhead *Sphyrna lewini* | 15 | 239 | 154–224 | 21.2–26.4 | 0.05–0.18 | 22.00–04.00 |

in temperature or weather patterns from climate change (e.g. [13,14]), destruction of habitats (e.g. [15]), the introduction of alien predators (e.g. [16,17]), overexploitation of prey resources (e.g. [18]) or depletion of predator populations (e.g. [19,20]) may drive predators to occupy or forage in new areas, potentially changing the assemblage of species co-occurring within a predator guild. Over the last decade, studies have started to unravel the partitioning mechanisms of some terrestrial predator guilds, such as the intact African large carnivore guild (e.g. [21,22]). However, the few studies that have examined the mechanisms allowing coexistence of large marine predator guilds (including in elasmobranchs and seabirds) have generally focused on resource-level partitioning (e.g. [23,24]), spatial partitioning (e.g. [24–27]) or seasonal partitioning where allopatric predators partition occupancy of an area on an annual basis (e.g. [28–30]). In particular, the occurrence of diel temporal partitioning of sympatric marine predators is poorly studied (although see [26]), probably due to the difficulty in determining diel foraging patterns in highly mobile underwater species.

The present study examined diel activity patterns and the potential for diel temporal niche partitioning in a guild of large coastal sharks. Activity patterns of six shark species in the Gulf of Mexico (Florida, USA) were determined using accelerometers deployed on free-ranging sharks. Because these six species seasonally co-occur (all present during winter months) [31] and show overlap in their prey as well as evidence of hierarchical predation within the guild (larger species are known to prey on some smaller species; [32,33] and references therein), we hypothesized that there would be a degree of diel temporal niche partitioning to limit interspecific competition and promote coexistence of all species.

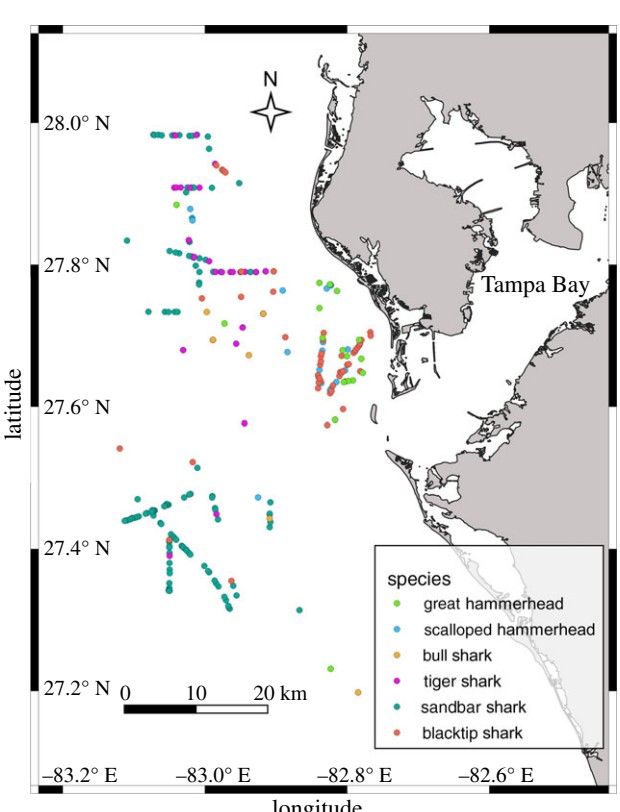

**Figure 1.** Study site and capture locations. Map of capture locations of large coastal sharks caught during winter months (November–April; water temperature approx. 16–26°C) in the current study. Capture locations for all individuals caught (tagged and untagged) are shown as an indication of their spatial distribution within the study area. (Online version in colour.)

## 2. Results

Sufficient accelerometer data (more than 5 individuals each tracked for more than 24 h) for analyses of activity patterns were obtained from six sympatric species of large coastal sharks: blacktip sharks (*Carcharhinus limbatus*), bull sharks (*Carcharhinus leucas*), sandbar sharks (*Carcharhinus plumbeus*), tiger sharks (*Galeocerdo cuvier*), great hammerhead sharks (*Sphyrna mokarran*) and scalloped hammerhead sharks (*Sphyrna lewini*) (table 1). In total, 3766 h of acceleration data from 172 individuals were used in our analyses (table 1). All

species were caught in winter months (November–April), at water temperatures spanning approximately 16–26°C (table 1 and figure 1). Tiger sharks from a large size range were caught and tagged, but smaller animals (less than 150 cm total length, TL) showed different activity patterns compared to larger animals (greater than 150 cm TL). Few small tiger sharks were tagged (*n* = 3), and therefore robust analysis of activity patterns was not feasible for this group. As a result, small individuals were removed from analysis and activity patterns of only larger sharks (greater than 150 cm TL) were included in the analyses.

Proc. R. Soc. B 288: 20210816

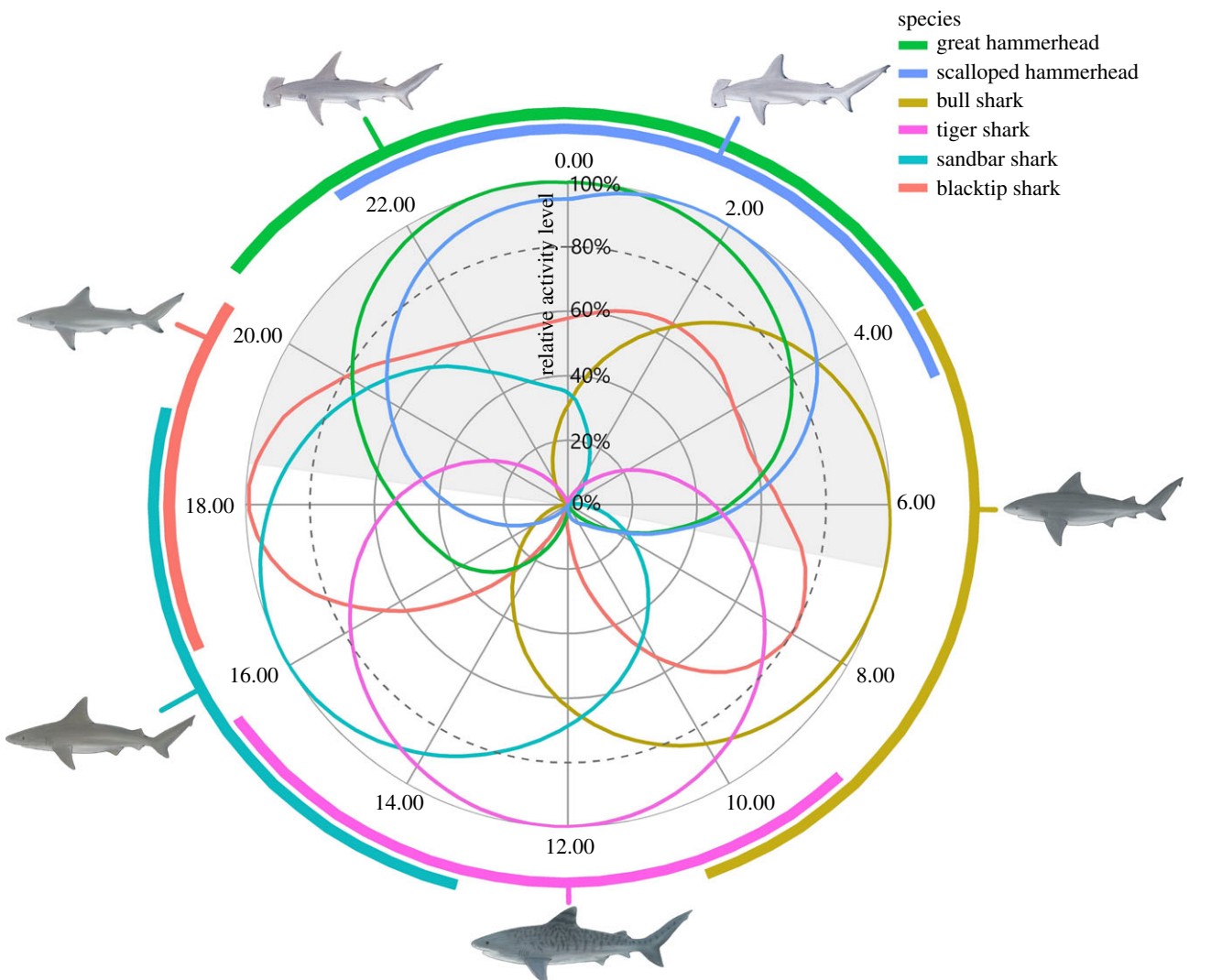

**Figure 2.** Diel activity patterns of co-occurring shark species. Diel activity patterns of six species of co-occurring large coastal sharks found in the Eastern Gulf of Mexico, Florida, USA. The shaded region indicates the approximate night-time period. Because of different levels and degrees of change of overall dynamic body acceleration (ODBA) recorded for different species, activity patterns are plotted here as a percentage of the difference between the minimum (0%) and maximum (100%) ODBA level recorded for each species. The coloured bars in the outer circle show the time span of peak activity (greater than or equal to 80% of maximum activity) of each species, with this 80% threshold indicated with a dotted line on the figure. For individual species trends, error and hourly data, see electronic supplementary material, figure S1. (Online version in colour.)

**Table 2.** Generalized additive mixed model selection table. Model selection table for generalized additive mixed models (GAMMs) used to determine diel patterns of activity. The top five models (based on the corrected Akaike's information criterion; AICc) are shown, with the best-fit model in bold. A single model was used for all species, with species included as a factor (×) in the smoother (denoted by 's()') with hour of day (HOD). Temp, temperature; TL, total length; DH, hour of deployment.

| model formula | AICc | ΔAICc | d.f. | log likelihood | $R^2$ |
|---|---|---|---|---|---|
| ODBA ∼ s(HOD × species) + Temp + TL | **−11143.9** | — | **22** | **5585.1** | **0.11** |
| ODBA ∼ s(HOD × species) + Temp + TL + DH | −11138.5 | 5.4 | 23 | 5583.4 | 0.12 |
| ODBA ∼ s(HOD × species) + Temp | −11134.7 | 9.2 | 21 | 5579.4 | 0.11 |
| ODBA ∼ s(HOD × species) + TL | −11134.4 | 9.5 | 21 | 5579.3 | 0.02 |
| ODBA ∼ s(HOD × species) + Temp + DH | −11133.1 | 10.8 | 22 | 5579.6 | 0.12 |

Generalized additive mixed models (GAMMs) were used to examine diel activity patterns. The top GAMM included time of day as a smoothed term, species as a factor within the smoother, and temperature and TL as fixed predictors (table 2). Temperature and TL influenced the intercept of the models, with higher overall dynamic body acceleration (ODBA) values generally observed at higher temperatures

and in smaller sharks (see electronic supplementary material, figure S1). The exclusion of hour of deployment as either a fixed predictor or smoothed term in the model confirms that results were not affected by any extended recovery of animals throughout the deployment. Each of the six large coastal shark species showed a different diel activity pattern with a limited overlap of peak activity times (figure 2), suggesting that

temporal partitioning is occurring. Bull sharks were most active in early morning hours, tiger sharks during midday, sandbar sharks during the afternoon, blacktip sharks during evening hours and both scalloped and great hammerhead sharks during night-time hours, the only two species with substantial overlap in timing of peak activity (figure 2 and table 1). The hourly mean acceleration values recorded for individuals of each species alongside the best-fit GAMM trend are plotted in electronic supplementary material, figure S1.

## 3. Discussion

The minimal overlap in diel timing of peak activity in the six large coastal shark species examined (with the exception of the two hammerhead species) provides evidence for the occurrence of temporal partitioning. To our knowledge, these results are the first example of diel temporal partitioning in a marine predator guild. Such partitioning is likely to be driven by a combination of physiological and morphological constraints of each species and behavioural mechanisms, including a species's potential for behavioural plasticity.

The six species examined here have been concurrently captured in the present study and in past work (e.g. [31]). It is also of note that in past studies and in the present study spinner sharks (*Carcharhinus brevipinna*; similar trophic position to blacktip sharks) were commonly caught alongside the six species examined here and may play a role in temporal partitioning patterns, although we obtained insufficient accelerometer data from this species to be included in the analyses. Past studies have defined most of the species studied here as generalist teleost/elasmobranch predators with dietary overlap (e.g. [34–37]), suggesting that there is limited spatial or resource-level partitioning. Thus it is perhaps not surprising that temporal partitioning exists, although it has been found to be rare in predator guilds compared with spatial or resource partitioning [6,38]. There is also the potential that a degree of spatial or seasonal temporal partitioning exists within this guild. For example, within the study area, sandbar sharks were typically caught further offshore, and blacktip, great hammerhead and scalloped hammerhead sharks more often caught close to shore (figure 1); and tiger sharks were more commonly caught in the lower half of the study temperature range, while great hammerheads were more common in the upper half of the range (table 1). However, all six species were caught within the same longline set (i.e. within approx. 10 km of each other at the same time) at some point during the study. Furthermore, all species studied here range widely throughout the same general area, suggesting they display considerable spatial overlap. It is also notable that a degree of resource partitioning may also exist and may result directly from temporal partitioning, as the assemblage of available prey species probably varies throughout the diel cycle [39,40]. Diet analyses have not been conducted on the sub-adult/adult populations of these species found within the specific area of the study; however, past work in nearby locations has determined that all species here are generalist teleost, elasmobranch and cephalopod predators [32,33,41–43]. The only notable differences in diets between species in past studies are that marine reptiles, seabirds and invertebrates (incl. molluscs and bivalves), in addition to teleosts and elasmobranchs, may be important parts of the diet of tiger sharks [44].

There are two main factors identified as potential reasons that temporal partitioning is rare compared with resource or spatial partitioning. First, circadian rhythms are often assumed to be heavily evolutionarily constrained [6,45–47], limiting the potential for co-occurring species to change their activity patterns to promote diel partitioning. However, recent studies have shown that within the bounds of evolutionary adaptations to diurnal rhythmicity (e.g. eye types or neurological adaptations that promote vision in either dark or light) many animals, including sharks, have the capacity to behaviourally shift their diel activity patterns in reaction to shifts in environmental variables, prey assemblages or the introduction/removal of a predator or competitor [7,8,48–51]. Second, if diel rhythmicity is constrained, temporal partitioning may represent a greater drop in potential energy acquisition compared with other partitioning mechanisms; if animals are unable to shift their diel activity patterns, they would partition time instead by ceasing to forage during certain times of the day, trading potential energy acquisition for no energy acquisition [1,6]. Conversely, with spatial or resource partitioning, if animals shift from optimal foraging conditions to foraging in a suboptimal habitat or on a suboptimal prey source, they theoretically trade optimal energy intake for lower energy intake. Because of the relatively large drop in potential energy intake theorized from temporal partitioning, previous work has suggested that for this type of partitioning to occur either competition for resources must be severe, or interference competition is present (often assumed more likely), where hierarchical predation within the predator guild exists and predators partition time to avoid agonistic attacks [6,52,53]. Elasmobranchs (including large sharks) have been identified as a main prey source for great hammerhead (but not scalloped hammerhead) sharks [37,54], tiger sharks in Hawaii and South Africa [55,56], and bull sharks in South Africa [57]. Elasmobranchs (but mainly rays or small sharks) have also been identified as an infrequent prey source for sandbar sharks [36,58] and blacktip sharks [34]. Therefore, interference competition may be a significant driver of activity patterns and behaviour in less dominant shark species. This may be particularly relevant for blacktip sharks, the smallest species examined here and most likely to be predated upon by larger sharks. By reducing activity at times where higher-order predators are most active, competitors which are lower on the trophic scale (i.e. blacktip sharks) can decrease their probability of detection by a predator at those times [59,60]. It is also possible that refuging and foraging occur in slightly different microhabitats, and that temporal partitioning of foraging times results in spatial separation of lower trophic order competitors that are also prey sources for more dominant species.

The diel temporal partitioning observed here in large coastal sharks is likely to be the result of a combination of constraints imposed by morphological or physiological limitations as well as behavioural plasticity. The larger or more dominant predators, including tiger sharks, bull sharks and great hammerhead sharks, may be active and forage during the times of day that best suit them physiologically. For example, hammerhead sharks (most active here at night) are known to have superior binocular vision compared to carcharhinid species which may put them at an advantage in low light environments [61], while tiger sharks (most active here during midday) have been proposed to use visual silhouettes of prey on the surface as a main foraging mechanism [62,63], requiring higher light levels. Conversely, the less dominant species such as blacktip sharks may shift their diel foraging patterns in order

to avoid interactions with larger species. This is supported by previous studies that show similar diurnal activity in tiger sharks and nocturnal activity in hammerhead sharks across different oceans and populations [31,49,64–67]. The maintenance of foraging rhythms across populations of hammerhead and tiger sharks experiencing different prey species, competitors and environmental conditions suggests that these foraging times are optimal for these species and are potentially highly regulated by conserved physiological mechanisms.

Conversely, diel rhythms of blacktip sharks (the lowest trophic level of sharks in the current study) have varied across populations, with previous work indicating that blacktip sharks forage at night [68,69] or during the morning [31], while here blacktip sharks were most active during the evening. When the same methods used in this study are applied to accelerometer data collected from blacktip sharks using the same area during summer months ($n = 12$) and in Florida Bay in October ($n = 9$), peaks of activity occur in the early morning and midday, respectively (N.M.W., K.O.L. & J.J.M. 2014–2017, unpublished data). This indicates a high degree of plasticity in diel rhythmicity in this species even over relatively small spatial or seasonal scales, most likely in response to shifting assemblages of predator, prey or competitor species. Such plasticity may be key in allowing this species to succeed in a variety of environments, and may provide a stronger buffer for deleterious trophic impacts stemming from environmental change compared to species with stricter diel patterns. Plastic diel rhythms may be especially important for mesopredator species in mid or low trophic levels, as these animals would be more likely to shift diel patterns as a result of interference competition. Whether diel foraging and activity patterns in specific species are driven more by morphological constraints or behavioural shifts requires further investigation, for example by examining diel activity patterns in assemblages of similar species across time and space.

## 4. Conclusions

Regardless of the relative contribution of morphological versus behavioural mechanisms in driving patterns of diel activity, this study has demonstrated the importance of diel temporal variability as a niche partitioning axis in a marine predator guild for the first time. Recent and historical work suggests that healthy marine ecosystems are almost always characterized by abundant and diverse predator populations, although such healthy or pristine systems are unfortunately becoming increasingly rare [70,71]. Considering that many species of sharks tend to share space with other species as both juveniles and adults, and often display an overlap in prey and hierarchical predation that would incite interference competition, diel temporal niche partitioning may be an important factor in allowing multiple shark species to coexist in a variety of ecosystems. Documenting the occurrence of temporal niche partitioning and determining the mechanistic drivers and interspecific relationships of such partitioning are essential to understanding the community ecology of marine ecosystems and to forecasting the direct and indirect repercussions of environmental change. This is particularly important considering that many predator and prey species in marine systems are exploited for human consumption, and rapid environmental change such as temperature shifts and habitat destruction may alter the assemblage of predators

or competitors present. Understanding the mechanisms that allow marine predators to coexist will help to preserve and restore healthy, predator-rich marine systems.

## 5. Methods

### (a) Animal capture and tagging

Large coastal sharks were caught in the Gulf of Mexico in open coastal waters near Madeira Beach, FL, using bottom longlines. Longline sets were deployed between 2013 and 2017, using 18/0 circle hooks baited with local teleost species including bonito (Scombridae) and ladyfish (*Elops saurus*), and with soak times ranging from 2 to 15 h. Sharks were briefly restrained onboard the vessel and gills irrigated with seawater during the tagging process. Sharks were measured (total length, TL), sexed and tagged with an acceleration data logger (model G6A+, Cefas, Lowestoft, UK), which recorded triaxial acceleration at 25 Hz, depth at 1 Hz and water temperature at 0.03 Hz. These data loggers were built into positively buoyant tag packages (see [72]), which were secured to the first dorsal fin of sharks with an attachment incorporating a galvanic timed release, set to dissolve and break the attachment at a predetermined time. Once tagging was complete, sharks were released at the site of capture. The tagging and measurement process took approximately 3–6 min. The galvanic timed releases were set to release the tag package after approximately 1–3 days, allowing the tag package to float to the surface. Tag packages were relocated using VHF telemetry (M130b transmitter and R410 receiver with 3-element yagi antenna; Advanced Telemetry Systems, Isanti, MN, USA) and physically recovered from a vessel following methods described by Lear & Whitney [73].

### (b) Data processing and analysis

Once acceleration data logger packages were recovered, the data were downloaded and analysed using Igor Pro (v. 6.8, Wavemetrics, Lake Oswego, OR, USA) and R (v. 3.3.1, R Foundation for Statistical Computing, Vienna, Austria). Data from the first 12 h after tagging were excluded to prevent biasing analyses with behavioural effects of the capture and tagging process [74]. Static acceleration was separated from dynamic acceleration using a 3 s box smoother, which was sufficient to remove tailbeat signals from the static acceleration for all species. The activity was calculated as ODBA, the sum of the absolute value of the dynamic acceleration from all three axes [75], which has widely been used as a measurement of activity in fish and other taxa (e.g. [76–78]) as it reflects the total movement of the animal in three dimensions.

To determine diel patterns of activity, mean ODBA was calculated for each hour of the deployment for each individual. Subsequently, a GAMM was constructed using the 'mgcv' package [79] in R (v. 1.8–12; R Foundation for Statistical Computing, Vienna, Austria). This model predicted activity (as ODBA) by hour of the day as a cyclic smoothed term, with species included as an interaction within the smoother. It is well established that temperature influences activity levels in ectothermic animals [78,80], and that ODBA and tailbeat frequency scale with body size [75,81,82]. Therefore, the total length of sharks and water temperature were also included as fixed predictors of activity in the GAMM. Additionally, the hour of the deployment was included as either a fixed predictor (allowing a change in the magnitude of activity) or a smoothed term (allowing change in the pattern of activity) to ensure that no effects of tagging or extended recovery biased results. Inclusion of these fixed predictors and smoothed terms in the final model was tested using the Akaike information criterion corrected for small sample size (AICc), with fixed predictors maintained if their inclusion in the GAMM produced models with an AICc > 2 lower than

models without the predictors included. The individual was included in all models as a random effect. Serial dependence of data was accounted for in the GAMMs by including an auto-regressive process of order 1 [83] using the CorAR1 function. Following the establishment of activity patterns, the most active period of the day for each species was identified as the time where activity was greater than or equal to 80% of the difference between the maximum and minimum activity levels observed for the species. This peak activity time was assumed to represent peak foraging time based on previous studies in similar animals (e.g. [59,84–88]).

Ethics. All work with animals was approved by Mote Marine Laboratory Institutional Animal Care and Use Committee (protocol no. 13-11-NW2).
Data accessibility. The processed hourly acceleration and related data for individuals used in this study are available in the electronic supplementary material [89].

Authors' contributions. K.O.L.: conceptualization, data curation, formal analysis, methodology, writing—original draft, writing—review and editing; N.M.W.: data curation, funding acquisition, writing—review and editing; J.J.M.: data curation, writing—review and editing; A.C.G.: conceptualization, formal analysis, writing—review and editing.
All authors gave final approval for publication and agreed to be held accountable for the work performed therein.

Competing interests. The authors declare no conflict of interests.
Funding. Funding for this study was provided by the National Oceanic and Atmospheric Administration Bycatch Reduction and Cooperative Research Programmes.
Acknowledgements. We are grateful to many people who assisted with the collection of data for this study, including commercial fishing captains R. Lauser, L. Hill, D. Campo, and J. Bonnell, captains D. Dougherty, G. Byrd, C. Jelicks, P. Hull and R. Kane, and pilots F. Casey, G. Roam and B. Powell who assisted with tag recovery, and numerous Mote Marine Laboratory staff and volunteers who assisted with capture and tagging, including H. Marshall, A. Andres, A. Ontkos and R. Hueter.

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
