## [Peer Review File · Proceedings of the Royal Society B: Biological Sciences]

Review History

RSPB-2021-0816.R0 (Original submission)

Review form: Reviewer 1 (Ross Dwyer)

Recommendation

Major revision is needed (please make suggestions in comments)

Scientific importance: Is the manuscript an original and important contribution to its field?

Excellent

General interest: Is the paper of sufficient general interest?

Excellent

Quality of the paper: Is the overall quality of the paper suitable?

Excellent

Is the length of the paper justified?

Yes

Should the paper be seen by a specialist statistical reviewer?

No

Do you have any concerns about statistical analyses in this paper? If so, please specify them explicitly in your report.

No

It is a condition of publication that authors make their supporting data, code and materials available - either as supplementary material or hosted in an external repository. Please rate, if applicable, the supporting data on the following criteria.

Is it accessible?

Yes

Is it clear?

Yes

Is it adequate?

Yes

Do you have any ethical concerns with this paper?

No

Comments to the Author

This is an interesting paper investigating diel niche partitioning amongst 6 cooccurring large-bodied coastal shark species. Historically, investigation into inter-specific variation in habitat use and diet in marine species are commonly derived from fisheries or trawl surveys, broad scale telemetric studies (e.g. passive acoustic telemetry), and/or inferred from tissue stable isotope analysis. Using a novel approach (based on ODBA values generated from onboard triaxial accelerometers) and an impressive multispecies dataset, the manuscript's authors were able to collect fine-scale behavioural information from a large sample of sharks that could be used to quantify changes in swimming activity over a 24-h period. Using a generalised additive mixed modelling approach, the authors found marked interspecific differences in the time of day when tagged sharks were at their most active - where only scalloped and great hammerhead sharks had a high degree of temporal overlap in their peak activity periods. The authors conclude that the observed differences in activity across the 6 species were most likely a result of interspecific competition and the response of predation pressure/interference competition from larger-bodied sharks.

In this manuscript, the study's design, figures, analysis and writing are all of high quality, with clearly stated hypotheses. The authors also introduce an impressive multispecies dataset with which to investigate temporal niche partitioning amongst number of active, wide ranging coastal sharks - this is to be commended as these ecological traits often make such comparisons difficult in this taxon.

While the paper has potential, there are several key issues and a series of specific comments that are listed below that the authors should address prior to publication.

1) In the Introduction (Line 42), the authors state "Theoretically, all partitioning mechanisms are a result of co-evolution, where interspecific competition has driven sympatric species to adapt specializations that allow them to co-occur [1-3]". Here the authors cite (amongst others) Connell (1980), however Connell (1980) actually states "the notion of coevolutionary shaping of competitors' niches has little support at present. Theory and evidence suggest that it is probable only in low diversity communities". The authors should consider reframing their introduction to avoid arguing that they have demonstrated any coevolutionary specialisation in this study (which, as the Connell paper argues, is challenging to demonstrate). The authors come to a logical conclusion about the mechanisms that may be shaping the patterns observed, however it should be made clear that this does not provide evidence for coevolved niche

specialisation.

2) The discussion focuses a great deal on both “evolutionary and behavioural mechanisms” (L 248-249), however animal behaviour (and its degree of plasticity) are also evolved traits. Do the authors mean “physiological constraints”, “morphological constraints”, and/or “behavioural plasticity”? This is important to clarify, as evidence of behavioural plasticity in different populations of the same species is one of the factors Connell (1980) explicitly discusses as a good example of what is not coevolved niche partitioning.

3) The paper makes several conclusions about this study being the first example of temporal niche partitioning in “large marine predators” (E.g. L23-24, L283-284). However, there are a number of previously published studies on temporal niche partitioning in large marine species (including sharks). For example, seasonal temporal niche partitioning has been shown previously in euryhaline sharks (Dwyer et al. 2020), and is widely known to occur in breeding seabirds (e.g. Granroth-Wilding and Phillips 2019; Monteiro and Furness 1998; Friesen et al 2007; Navarro et al. 2013)). While examples of “diel temporal partitioning” are less common in the literature (particularly using large multispecies activity datasets such as this), the authors should clarify that diel temporal partitioning is specifically what they are quantifying.

Dwyer, RG, Campbell, HA, Cramp, RL, et al. (2020) Niche partitioning between river shark species is driven by seasonal fluctuations in environmental salinity. *Functional Ecology*; 34: 2170–2185.

Granroth-Wilding, H.M.V. and Phillips, R.A. (2019), Segregation in space and time explains the coexistence of two sympatric sub-Antarctic petrels. *Ibis*, 161: 101-116.
<https://doi.org/10.1111/ibi.12584>

Monteiro L. R. and Furness R. W. (1998) Speciation through temporal segregation of Madeiran storm petrel (*Oceanodroma castro*) populations in the Azores? *Phil. Trans. R. Soc. Lond. B* 353 945–953

V. L. Friesen, A. L. Smith, E. Gómez-Díaz, M. Bolton, R. W. Furness, J. González-Solís, L. R. Monteiro (2007) Sympatric speciation by allochrony in a seabird *Proceedings of the National Academy of Sciences* Nov 2007, 104 (47) 18589-18594; DOI: 10.1073/pnas.0700446104

Navarro J, Votier SC, Aguzzi J, Chiesa JJ, Forero MG, Phillips RA (2013) Ecological Segregation in Space, Time and Trophic Niche of Sympatric Planktivorous Petrels. *PLoS ONE* 8(4): e62897.
<https://doi.org/10.1371/journal.pone.0062897>

4) One of the main assumptions of this work is that these were coinhabiting sharks that shared the same geographic space, but partitioned their peak activities around a 24-h cycle. As such, I would like to see evidence of where these animals were captured and the dataloggers retrieved as some sort of figure. There also appears to be no temperature overlap between tiger sharks and great hammerhead sharks in table 1. Does this suggest that these species are using different areas of the study site/water column? If so this might suggest some form of spatial partitioning between species was also present.

Specific comments

Line 24: Suggest “diel”, but not “temporal”: there are certainly many descriptions of temporal niche partitioning in seabirds (also a marine predator) in terms of offsetting breeding times (see above citations) and seasonal niche partitioning has been previously observed in one of the study species (i.e. bull sharks) – see above citations.

L 56-59: references would be useful here.

L 70-74: references would be useful here.

Lines 79-80. A map of the tag release and package recovery locations would add extra weight to the idea that these animals were using the same 'space' but shifting their activity throughout the day to avoid peak foraging times (Schoener 1974; Humphries et al. 2016). Even if this map is put in the appendices, I feel it would be useful for readers.

L 115-116: confusing sentence that needs rephrasing, also missing full stop.

L117-118. Why include the hour of the deployment "as either a fixed predictor OR a smoothed term". What was the rationale for including it as both as a fixed and as a random term?

Figure 1. this is a nicely crafted figure, but it is rather difficult to extract the shape of the data for all six species. There is also no indication of the error associated with each of these predicted lines. Perhaps keep the larger image but also provide 6 smaller images (as part of Figure 1 or in the supplementary files) showing the % activity patterns for each of the 6 species, and the 95% confidence intervals of the gam models. Presenting the mean model predictions does not provide insight into within-species variability, which is also important.

Line 131. Why did you select >5 individuals per species as a cut off? From looking at table 1 the minimum number of individuals for a species is 11 (bull sharks). Was this to do with the available degrees of freedom for your model or some other factor?

Line 161-167. In Table 1, there appears to be no temperature overlap between tiger sharks and great hammerhead sharks. Why this might be? Does this suggest that different areas of the study site/water column are used by these different species, suggesting spatial segregation as well as diel segregation?

Lines 170-175 (Table 2). Given that Temperature and TL both came up as significant in the GAMM, why not provide some insight into how these continuous (?) variables affected ODBA for each species? It would be nice to see some kind of visualisation / output of model smooths over the real data of their best model which captures the relationship between predictor and response variables.

Line 194. In paragraph 1 of the introduction, the authors use the definition of Schoener (1974) where temporal partitioning is where sympatric species rotate peak foraging times on a diel or seasonal scale. The authors go on to define that they are particularly interested in diel temporal partitioning of cooccurring marine predators where they are looking at changes of activity around a 24-h cycle (lines 63-65). As such I would suggest that you stick with the term diel temporal partitioning as this distinguishes this study from other examples in the published literature that have previously described seasonal niche partitioning in sharks (Dwyer et al 2020) and other kinds of temporal foraging segregation described in many species of seabirds (some references suggested above in comment for L 24).

L 234: I find this a compelling discussion point, however surely this then indicates the difference in timing of foraging activity is not resource partitioning at all, but simply predator avoidance behaviour? This comes back to what I think is the misinterpretation of the Connell paper - not all observed "partitioning" is coevolved specialisation, some is a result of plastic adaptation to current conditions.

L 243: I don't think reference 8 supports this contention.

L 248-249: But both behaviour (and its plasticity) are evolved traits? I see what you're saying (L 249-254) but this needs to be made clearer. In addition, if the less dominant sharks would immediately change their behaviour in the absence of the dominant predators, then this is not

coevolved niche partitioning but behavioural plasticity.

L 278: As discussed in main comments, be clear whether you mean “physiological constraints”, or “morphological constraints” instead of “evolutionary constraints”? Again, behaviour and its plasticity are evolved.

Line 283-284. As before I would suggest stating that this study specifically investigated diel temporal partitioning highlights the novelty of the findings, as there is a great deal of evidence for other forms of temporal partitioning in other taxa.

Review form: Reviewer 2

Recommendation

Major revision is needed (please make suggestions in comments)

Scientific importance: Is the manuscript an original and important contribution to its field?

Acceptable

General interest: Is the paper of sufficient general interest?

Acceptable

Quality of the paper: Is the overall quality of the paper suitable?

Acceptable

Is the length of the paper justified?

Yes

Should the paper be seen by a specialist statistical reviewer?

No

Do you have any concerns about statistical analyses in this paper? If so, please specify them explicitly in your report.

No

It is a condition of publication that authors make their supporting data, code and materials available - either as supplementary material or hosted in an external repository. Please rate, if applicable, the supporting data on the following criteria.

Is it accessible?

N/A

Is it clear?

N/A

Is it adequate?

N/A

Do you have any ethical concerns with this paper?

No

Comments to the Author

The manuscript include a novel technique to follow sharks to know the time used in the water and probaly sharing resources.

I have some questions.

L 133 the name species of bull shark is *C. leucas*.

L148 to 152. authors explain the time used by different sharks; however is very important to know what kind of prey they consume to know the shark species trophic habitat (bottom, pelagic, mesopelagic). Then authors must include background about the feeding habits of this shark species using published information. mainly from the study area. as an example *S. lewini* in the Gulf of California feed on benthic and coastal prey when they are juveniles (mainly fishes during the night), but when they are adults feed on squids in oceanic waters. Also *C. limbatus* predate mainly on fishes (sardines) in epipelagic coastal waters. Then that is the importance to give more information about the prey of these shark species in the study area to compare with the movements in the accelerometers.

Decision letter (RSPB-2021-0816.R0)

05-May-2021

Dear Dr Lear:

Your manuscript has now been peer reviewed and the reviews have been assessed by an Associate Editor. The reviewers' comments (not including confidential comments to the Editor) and the comments from the Associate Editor are included at the end of this email for your reference. As you will see, the reviewers and the Editors have raised some concerns with your manuscript and we would like to invite you to revise your manuscript to address them.

Research ethics:

Use of animals and field studies:

It is a condition of publication that you make available the data and research materials supporting the results in the article. Please see our Data Sharing Policies (<https://royalsociety.org/journals/authors/author-guidelines/#data>). Datasets should be deposited in an appropriate publicly available repository and details of the associated accession number, link or DOI to the datasets must be included in the Data Accessibility section of the article (<https://royalsociety.org/journals/ethics-policies/data-sharing-mining/>). Reference(s) to datasets should also be included in the reference list of the article with DOIs (where available).

Please submit a copy of your revised paper within three weeks. If we do not hear from you within this time your manuscript will be rejected. If you are unable to meet this deadline please let us know as soon as possible, as we may be able to grant a short extension.

Best wishes,

Dr Daniel Costa

Associate Editor

Board Member: 1

Comments to Author:

Both reviewers have identified that this is an interesting manuscript worth publishing but have identified issues that would need addressing before the manuscript could proceed to publication. Please note the reviewers comments in revising the manuscript, particularly concerns around stretching inferences beyond what the data can support and concerns around spatial partitioning not currently adequately addressed at present by the manuscript.

Reviewer(s)' Comments to Author:

Referee: 1

Comments to the Author(s)

This is an interesting paper investigating diel niche partitioning amongst 6 cooccurring large-bodied coastal shark species. Historically, investigation into inter-specific variation in habitat use and diet in marine species are commonly derived from fisheries or trawl surveys, broad scale telemetric studies (e.g. passive acoustic telemetry), and/or inferred from tissue stable isotope analysis. Using a novel approach (based on ODBA values generated from onboard triaxial accelerometers) and an impressive multispecies dataset, the manuscript's authors were able to collect fine-scale behavioural information from a large sample of sharks that could be used to quantify changes in swimming activity over a 24-h period. Using a generalised additive mixed modelling approach, the authors found marked interspecific differences in the time of day when tagged sharks were at their most active - where only scalloped and great hammerhead sharks had a high degree of temporal overlap in their peak activity periods. The authors conclude that the observed differences in activity across the 6 species were most likely a result of interspecific competition and the response of predation pressure/interference competition from larger-bodied sharks.

In this manuscript, the study's design, figures, analysis and writing are all of high quality, with clearly stated hypotheses. The authors also introduce an impressive multispecies dataset with which to investigate temporal niche partitioning amongst number of active, wide ranging coastal sharks - this is to be commended as these ecological traits often make such comparisons difficult in this taxon.

While the paper has potential, there are several key issues and a series of specific comments that are listed below that the authors should address prior to publication.

1) In the Introduction (Line 42), the authors state "Theoretically, all partitioning mechanisms are a result of co-evolution, where interspecific competition has driven sympatric species to adapt specializations that allow them to co-occur [1-3]". Here the authors cite (amongst others) Connell (1980), however Connell (1980) actually states "the notion of coevolutionary shaping of competitors' niches has little support at present. Theory and evidence suggest that it is probable only in low diversity communities". The authors should consider reframing their introduction to avoid arguing that they have demonstrated any coevolutionary specialisation in this study (which, as the Connell paper argues, is challenging to demonstrate). The authors come to a logical conclusion about the mechanisms that may be shaping the patterns observed, however it should be made clear that this does not provide evidence for coevolved niche specialisation.

2) The discussion focuses a great deal on both "evolutionary and behavioural mechanisms" (L 248-249), however animal behaviour (and its degree of plasticity) are also evolved traits. Do the authors mean "physiological constraints", "morphological constraints", and/or "behavioural plasticity"? This is important to clarify, as evidence of behavioural plasticity in different populations of the same species is one of the factors Connell (1980) explicitly discusses as a good example of what is not coevolved niche partitioning.

3) The paper makes several conclusions about this study being the first example of temporal niche partitioning in “large marine predators” (E.g. L23-24, L283-284). However, there are a number of previously published studies on temporal niche partitioning in large marine species (including sharks). For example, seasonal temporal niche partitioning has been shown previously in euryhaline sharks (Dwyer et al. 2020), and is widely known to occur in breeding seabirds (e.g. Granroth-Wilding and Phillips 2019; Monteiro and Furness 1998; Friesen et al 2007; Navarro et al. 2013)). While examples of “diel temporal partitioning” are less common in the literature (particularly using large multispecies activity datasets such as this), the authors should clarify that diel temporal partitioning is specifically what they are quantifying.

Dwyer, RG, Campbell, HA, Cramp, RL, et al. (2020) Niche partitioning between river shark species is driven by seasonal fluctuations in environmental salinity. *Functional Ecology*; 34: 2170–2185.

Granroth-Wilding, H.M.V. and Phillips, R.A. (2019), Segregation in space and time explains the coexistence of two sympatric sub-Antarctic petrels. *Ibis*, 161: 101-116.
<https://doi.org/10.1111/ibi.12584>

Monteiro L. R. and Furness R. W. (1998) Speciation through temporal segregation of Madeiran storm petrel (*Oceanodroma castro*) populations in the Azores? *Phil. Trans. R. Soc. Lond. B* 353 945–953

V. L. Friesen, A. L. Smith, E. Gómez-Díaz, M. Bolton, R. W. Furness, J. González-Solís, L. R. Monteiro (2007) Sympatric speciation by allochrony in a seabird *Proceedings of the National Academy of Sciences* Nov 2007, 104 (47) 18589-18594; DOI: 10.1073/pnas.0700446104

Navarro J, Votier SC, Aguzzi J, Chiesa JJ, Forero MG, Phillips RA (2013) Ecological Segregation in Space, Time and Trophic Niche of Sympatric Planktivorous Petrels. *PLoS ONE* 8(4): e62897.
<https://doi.org/10.1371/journal.pone.0062897>

4) One of the main assumptions of this work is that these were coinhabiting sharks that shared the same geographic space, but partitioned their peak activities around a 24-h cycle. As such, I would like to see evidence of where these animals were captured and the dataloggers retrieved as some sort of figure. There also appears to be no temperature overlap between tiger sharks and great hammerhead sharks in table 1. Does this suggest that these species are using different areas of the study site/water column? If so this might suggest some form of spatial partitioning between species was also present.

Specific comments

Line 24: Suggest “diel”, but not “temporal”: there are certainly many descriptions of temporal niche partitioning in seabirds (also a marine predator) in terms of offsetting breeding times (see above citations) and seasonal niche partitioning has been previously observed in one of the study species (i.e. bull sharks) – see above citations.

L 56-59: references would be useful here.

L 70-74: references would be useful here.

Lines 79-80. A map of the tag release and package recovery locations would add extra weight to the idea that these animals were using the same ‘space’ but shifting their activity throughout the day to avoid peak foraging times (Schoener 1974; Humphries et al. 2016). Even if this map is put in the appendices, I feel it would be useful for readers.

L 115-116: confusing sentence that needs rephrasing, also missing full stop.

L117-118. Why include the hour of the deployment “as either a fixed predictor OR a smoothed term”. What was the rationale for including it as both as a fixed and as a random term?

Figure 1. This is a nicely crafted figure, but it is rather difficult to extract the shape of the data for all six species. There is also no indication of the error associated with each of these predicted lines. Perhaps keep the larger image but also provide 6 smaller images (as part of Figure 1 or in the supplementary files) showing the % activity patterns for each of the 6 species, and the 95% confidence intervals of the GAM models. Presenting the mean model predictions does not provide insight into within-species variability, which is also important.

Line 131. Why did you select >5 individuals per species as a cut off? From looking at table 1 the minimum number of individuals for a species is 11 (bull sharks). Was this to do with the available degrees of freedom for your model or some other factor?

Line 161-167. In Table 1, there appears to be no temperature overlap between tiger sharks and great hammerhead sharks. Why this might be? Does this suggest that different areas of the study site/water column are used by these different species, suggesting spatial segregation as well as diel segregation?

Lines 170-175 (Table 2). Given that Temperature and TL both came up as significant in the GAMM, why not provide some insight into how these continuous (?) variables affected ODBA for each species? It would be nice to see some kind of visualisation / output of model smooths over the real data of their best model which captures the relationship between predictor and response variables.

Line 194. In paragraph 1 of the introduction, the authors use the definition of Schoener (1974) where temporal partitioning is where sympatric species rotate peak foraging times on a diel or seasonal scale. The authors go on to define that they are particularly interested in diel temporal partitioning of cooccurring marine predators where they are looking at changes of activity around a 24-h cycle (lines 63-65). As such I would suggest that you stick with the term diel temporal partitioning as this distinguishes this study from other examples in the published literature that have previously described seasonal niche partitioning in sharks (Dwyer et al 2020) and other kinds of temporal foraging segregation described in many species of seabirds (some references suggested above in comment for L 24).

L 234: I find this a compelling discussion point, however surely this then indicates the difference in timing of foraging activity is not resource partitioning at all, but simply predator avoidance behaviour? This comes back to what I think is the misinterpretation of the Connell paper - not all observed “partitioning” is coevolved specialisation, some is a result of plastic adaptation to current conditions.

L 243: I don't think reference 8 supports this contention.

L 248-249: But both behaviour (and its plasticity) are evolved traits? I see what you're saying (L 249-254) but this needs to be made clearer. In addition, if the less dominant sharks would immediately change their behaviour in the absence of the dominant predators, then this is not coevolved niche partitioning but behavioural plasticity.

L 278: As discussed in main comments, be clear whether you mean “physiological constraints”, or “morphological constraints” instead of “evolutionary constraints”? Again, behaviour and its plasticity are evolved.

Line 283-284. As before I would suggest stating that this study specifically investigated diel temporal partitioning highlights the novelty of the findings, as there is a great deal of evidence for other forms of temporal partitioning in other taxa.

Referee: 2

Comments to the Author(s)

The manuscript include a novel technique to follow sharks to know the time used in the water and probaly sharing resources.

I have some questions.

L 133 the name species of bull shark is *C. leucas*.

L148 to 152. authors explain the time used by different sharks; however is very important to know what kind of prey they consume to know the shark species trophic habitat (bottom, pelagic, mesopelagic). Then authors must include background about the feeding habits of this shark species using published information. mainly from the study area. as an example *S. lewini* in the Gulf of California feed on benthic and coastal prey when they are juveniles (maily fishes during the night), but when they are adults feed on squids in oceanic waters. Also *C. limbatus* predate mainly on fishes (sardines) in epipelagic coastal waters. Then that is the impiortance to give more information about the prey of these shark species in the study area to compare with the movements in the accelerometers.

Author's Response to Decision Letter for (RSPB-2021-0816.R0)

See Appendix A.

Decision letter (RSPB-2021-0816.R1)

09-Jun-2021

Dear Dr Lear

I am pleased to inform you that your Review manuscript RSPB-2021-0816.R1 entitled "Temporal niche partitioning as a novel mechanism promoting co-existence of sympatric predators in marine systems" has been accepted for publication in Proceedings B.

The referee(s) do not recommend any further changes. Therefore, please proof-read your manuscript carefully and upload your final files for publication. Because the schedule for publication is very tight, it is a condition of publication that you submit the revised version of your manuscript within 7 days. If you do not think you will be able to meet this date please let me know immediately.

To upload your manuscript, log into <http://mc.manuscriptcentral.com/prsb> and enter your Author Centre, where you will find your manuscript title listed under "Manuscripts with Decisions." Under "Actions," click on "Create a Revision." Your manuscript number has been appended to denote a revision.

You will be unable to make your revisions on the originally submitted version of the manuscript. Instead, upload a new version through your Author Centre.

1) A text file of the manuscript (doc, txt, rtf or tex), including the references, tables (including captions) and figure captions. Please remove any tracked changes from the text before submission. PDF files are not an accepted format for the "Main Document".

2) A separate electronic file of each figure (tiff, EPS or print-quality PDF preferred). The format should be produced directly from original creation package, or original software format. Please note that PowerPoint files are not accepted.

3) Electronic supplementary material: this should be contained in a separate file from the main text and the file name should contain the author's name and journal name, e.g. `authorname_procb_ESM_figures.pdf`

All supplementary materials accompanying an accepted article will be treated as in their final form. They will be published alongside the paper on the journal website and posted on the online figshare repository. Files on figshare will be made available approximately one week before the accompanying article so that the supplementary material can be attributed a unique DOI. Please see: <https://royalsociety.org/journals/authors/author-guidelines/>

4) Data-Sharing and data citation

It is a condition of publication that data supporting your paper are made available. Data should be made available either in the electronic supplementary material or through an appropriate repository. Details of how to access data should be included in your paper. Please see <https://royalsociety.org/journals/ethics-policies/data-sharing-mining/> for more details.

<http://datadryad.org/submit?journalID=RSPB&manu=RSPB-2021-0816.R1> which will take you to your unique entry in the Dryad repository.

Once again, thank you for submitting your manuscript to Proceedings B and I look forward to receiving your final version. If you have any questions at all, please do not hesitate to get in touch.

Sincerely,
Dr Daniel Costa
Editor, Proceedings B
<mailto:proceedingsb@royalsociety.org>

Associate Editor

Comments to Author:

Thankyou for addressing the reviewers comments. Please note some additional editorial corrections that are required (included in the attached). Please also include a data, materials and code section in the end statements, as per the guide to authors. Once these have been completed I will be happy to proceed your manuscript to publication

Decision letter (RSPB-2021-0816.R2)

14-Jun-2021

Dear Dr Lear

I am pleased to inform you that your manuscript entitled "Temporal niche partitioning as a novel mechanism promoting co-existence of sympatric predators in marine systems" has been accepted for publication in Proceedings B.

Your article has been estimated as being 9 pages long. Our Production Office will be able to confirm the exact length at proof stage.

Data Accessibility section

Open Access

Paper charges

Sincerely,

Proceedings B

Appendix A

Response to reviewer comments

MS ID RSPB-2021-0816

Temporal niche partitioning as a novel mechanism promoting co-existence of sympatric predators in marine systems

Karissa O. Lear, Nicholas M. Whitney, John J. Morris, Adrian C. Gleiss

We would like to thank both referees for their time and expertise in reviewing our manuscript and suggesting detailed changes that have increased the quality and clarity of the text. We have responded to all comments below.

Referee: 1

General comments:

This is an interesting paper investigating diel niche partitioning amongst 6 cooccurring large-bodied coastal shark species. Historically, investigation into inter-specific variation in habitat use and diet in marine species are commonly derived from fisheries or trawl surveys, broad scale telemetric studies (e.g. passive acoustic telemetry), and/or inferred from tissue stable isotope analysis. Using a novel approach (based on ODBA values generated from onboard triaxial accelerometers) and an impressive multispecies dataset, the manuscript's authors were able to collect fine-scale behavioural information from a large sample of sharks that could be used to quantify changes in swimming activity over a 24-h period. Using a generalised additive mixed modelling approach, the authors found marked interspecific differences in the time of day when tagged sharks were at their most active – where only scalloped and great hammerhead sharks had a high degree of temporal overlap in their peak activity periods. The authors conclude that the observed differences in activity across the 6 species were most likely a result of interspecific competition and the response of predation pressure/interference competition from larger-bodied sharks.

In this manuscript, the study's design, figures, analysis and writing are all of high quality, with clearly stated hypotheses. The authors also introduce an impressive multispecies dataset with which to investigate temporal niche partitioning amongst number of active, wide ranging coastal sharks – this is to be commended as these ecological traits often make such comparisons difficult in this taxon.

While the paper has potential, there are several key issues and a series of specific comments that are listed below that the authors should address prior to publication.

1) In the Introduction (Line 42), the authors state “Theoretically, all partitioning mechanisms are a result of co-evolution, where interspecific competition has driven sympatric species to adapt specializations that allow them to co-occur [1-3]”. Here the authors cite (amongst others) Connell (1980), however Connell (1980) actually states “the notion of coevolutionary shaping of competitors' niches has little support at present. Theory and evidence suggest that it is probable only in low

diversity communities”. The authors should consider reframing their introduction to avoid arguing that they have demonstrated any coevolutionary specialisation in this study (which, as the Connell paper argues, is challenging to demonstrate). The authors come to a logical conclusion about the mechanisms that may be shaping the patterns observed, however it should be made clear that this does not provide evidence for coevolved niche specialisation.

Thank you for the clarification here. We have rephrased the introduction to remove discussion of co-evolution of partitioning mechanisms and focus instead, as suggested elsewhere, on physiological/morphological constraints vs. behavioural plasticity.

2) The discussion focuses a great deal on both “evolutionary and behavioural mechanisms” (L 248-249), however animal behaviour (and its degree of plasticity) are also evolved traits. Do the authors mean “physiological constraints”, “morphological constraints”, and/or “behavioural plasticity”? This is important to clarify, as evidence of behavioural plasticity in different populations of the same species is one of the factors Connell (1980) explicitly discusses as a good example of what is not coevolved niche partitioning.

This is an excellent point, and our intention was to divide morphological/physiological vs. behavioural rather than ‘evolutionary’ vs. behavioural. We have altered the language throughout to specify this better.

3) The paper makes several conclusions about this study being the first example of temporal niche partitioning in “large marine predators” (E.g. L23-24, L283-284). However, there are a number of previously published studies on temporal niche partitioning in large marine species (including sharks). For example, seasonal temporal niche partitioning has been shown previously in euryhaline sharks (Dwyer et al. 2020), and is widely known to occur in breeding seabirds (e.g. Granroth-Wilding and Phillips 2019; Monteiro and Furness 1998; Friesen et al 2007; Navarro et al. 2013)). While examples of “diel temporal partitioning” are less common in the literature (particularly using large multispecies activity datasets such as this), the authors should clarify that diel temporal partitioning is specifically what they are quantifying.

Thank you, we have clarified throughout that we are specifically referring to diel temporal partitioning and discussed the difference with seasonal temporal partitioning in lines:

“However, the few studies that have examined the mechanisms allowing coexistence of large marine predator guilds including in elasmobranchs and seabirds have generally focused on resource level partitioning (e.g. Shipley et al. 2019; Papastamatiou et al., 2006), spatial partitioning (e.g. Dwyer et al., 2020; Humphries et al., 2016; Papastamatiou et al., 2006; Granroth-Wilding and Phillips 2019), or seasonal partitioning where allopatric predators partition occupancy of an area on an annual basis (e.g. Matich et al., 2017; Granroth-Wilding and Phillips 2019; Monteiro and Furness 1998). In particular, the occurrence of diel temporal partitioning of sympatric marine predators is poorly studied (although see Navarro et al., 2013), likely due to the difficulty in determining diel foraging patterns in highly mobile underwater species.”

4) One of the main assumptions of this work is that these were coinhabiting sharks that shared the

same geographic space, but partitioned their peak activities around a 24-h cycle. As such, I would like to see evidence of where these animals were captured and the dataloggers retrieved as some sort of figure. There also appears to be no temperature overlap between tiger sharks and great hammerhead sharks in table 1. Does this suggest that these species are using different areas of the study site/water column? If so this might suggest some form of spatial partitioning between species was also present.

We have now included a map of the capture locations of all sharks from the six species caught during the study, and have added a more thorough description of the potential degree of spatial partitioning in lines 207-215:

“There is also the potential that a degree of spatial or seasonal temporal partitioning exists within this guild. For example, within the study area, sandbar sharks were typically caught further offshore and blacktip, great hammerhead, and scalloped hammerhead sharks more often caught close to shore (see Fig. 1), and tiger sharks were more commonly caught in the lower half of the study temperature range while great hammerheads were more common in the upper half of the range (see Table 1). However, all six species were caught within the same longline set (i.e. within ~10 km of each other at the same time) as each other species analysed here at some point during the study and all range widely in the same general area, suggesting spatial overlap.”

Specific comments

#	Line	Referee Comment	Author response
1	24	Suggest “diel”, but not “temporal”: there are certainly many descriptions of temporal niche partitioning in seabirds (also a marine predator) in terms of offsetting breeding times (see above citations) and seasonal niche partitioning has been previously observed in one of the study species (i.e. bull sharks) – see above citations.	Thank you, this change has been made.
2	56-59	references would be useful here.	References have been added
3	70-74	references would be useful here	References have been added
4	79-80	A map of the tag release and package recovery locations would add extra weight to the idea that these animals were using the same ‘space’ but shifting their activity throughout the day to avoid peak foraging times (Schoener 1974; Humphries et al. 2016). Even if this map is put in the appendices, I feel it would be useful for readers	This has been added to the manuscript.

5	115-116	confusing sentence that needs rephrasing, also missing full stop.	This has been rephrased: "It is well established that temperature influences activity levels in ectothermic animals [27, 29], and that ODBA and tailbeat frequency scale with body size [24, 30, 31]. Therefore, total length of sharks and water temperature were also included as fixed predictors of activity in the GAMM."
6	117-118	Why include the hour of the deployment "as either a fixed predictor OR a smoothed term". What was the rationale for including it as both as a fixed and as a random term?	If the hour of deployment was significant as a fixed term, this would allow hour of deployment to change the intercept of the model (e.g. if ODBA decreased throughout the deployment, indicating that the shark was still recovering from capture during the measurement period). If included as a smoothed term, this allows hour of deployment to change the pattern of activity (e.g. if there was no activity pattern in the first part and then an activity pattern emerged as the shark recovered). We have now clarified this (lines 118-120): "Additionally, the hour of the deployment was included as either a fixed predictor (allowing change in the magnitude of activity) or a smoothed term (allowing change in the pattern of activity) to ensure that no effects of tagging or extended recovery biased results."
7	Figure 1.	this is a nicely crafted figure, but it is rather difficult to extract the shape of the data for all six species. There is also no indication of the error associated with each of these predicted lines. Perhaps keep the larger image but also provide 6 smaller images (as part of Figure 1 or in the supplementary files) showing the % activity patterns for each of the 6 species, and the 95% confidence intervals of the gam models. Presenting the mean model	Supplementary Figure S1 shows the individual trends of each species, plotted over raw data and with model error estimations shown. This figure is now referenced within the legend of Figure 2.

		predictions does not provide insight into within-species variability, which is also important.	
8	131	Why did you select >5 individuals per species as a cut off? From looking at table 1 the minimum number of individuals for a species is 11 (bull sharks). Was this to do with the available degrees of freedom for your model or some other factor?	Our cut-off of five individuals was based on previous experience using these types of data and in looking at high variation in trends in datasets (i.e. spinner sharks) with <5 individuals present – which did not have robust coverage of all points during the diel cycle.
9	161-167	In Table 1, there appears to be no temperature overlap between tiger sharks and great hammerhead sharks. Why this might be? Does this suggest that different areas of the study site/water column are used by these different species, suggesting spatial segregation as well as diel segregation?	This is mostly due to some amount of seasonal partitioning, which is referenced directly now in lines 212-219 (below). Tigers were more common in the colder months, while great hammerheads were more common in the warmer winter months, although tigers were caught all year round (but data from warmer months was not used because the tags were not on the sharks long enough to measure recovery), and both species were caught concurrently at 23.5 C – so there is some level of overlap. “There is also the potential that a degree of spatial or seasonal temporal partitioning exists within this guild. For example, within the study area, sandbar sharks were typically caught further offshore and blacktip, great hammerhead, and scalloped hammerhead sharks more often caught close to shore (see Fig. 1), and tiger sharks were more commonly caught in the lower half of the study temperature range while great hammerheads were more common in the upper half of the range (see Table 1). However, all six species were caught within the same longline set (i.e. within ~10 km of each other at the same time) as each other species analysed here at some point during the study and all range widely in the same general area, suggesting spatial overlap.”

10	170-175	(Table 2). Given that Temperature and TL both came up as significant in the GAMM, why not provide some insight into how these continuous (?) variables affected ODBA for each species? It would be nice to see some kind of visualisation / output of model smooths over the real data of their best model which captures the relationship between predictor and response variables.	A further description of these effects has been added to the text in lines 146-148, and the model outputs over individual shark data are supplied in Supplemental Figure S1, which shows general trends in total length. “Temperature and TL influenced the intercept of the models, with higher ODBA values generally observed at higher temperatures and in smaller sharks (see Supplementary information, Figure S1).”
11	194	In paragraph 1 of the introduction, the authors use the definition of Schoener (1974) where temporal partitioning is where sympatric species rotate peak foraging times on a diel or seasonal scale. The authors go on to define that they are particularly interested in diel temporal partitioning of cooccurring marine predators where they are looking at changes of activity around a 24-h cycle (lines 63-65). As such I would suggest that you stick with the term diel temporal partitioning as this distinguishes this study from other examples in the published literature that have previously described seasonal niche partitioning in sharks (Dwyer et al 2020) and other kinds of temporal foraging segregation described in many species of seabirds (some references suggested above in comment for L 24).	Thank you, we have specified that we are referring to diel temporal partitioning of sympatric sharks throughout.
12	234	I find this a compelling discussion point, however surely this then indicates the difference in timing of foraging activity is not resource partitioning at all, but simply predator avoidance behaviour? This comes back to what I think is the misinterpretation of the Connell paper - not all observed “partitioning” is coevolved specialisation, some is a result of plastic adaptation to current conditions.	We have changed our language throughout to not refer to co-evolution, but discuss behavioural plasticity of rhythms within the time allowed by morphological/physiological constraints. (e.g. see response to comment 14)

13	243	I don't think reference 8 supports this contention.	We have amended this sentence: "Another reason that temporal partitioning is presumed rare is that circadian rhythms are often assumed to be heavily evolutionarily constrained (Daan 1981; Kronfeld-Schor et al., 2001; Kronfeld-Schor and Dayan 2008), limiting the potential for co-occurring species to change their activity patterns to promote diel partitioning."
14	248-249	But both behaviour (and its plasticity) are evolved traits? I see what you're saying (L 249-254) but this needs to be made clearer. In addition, if the less dominant sharks would immediately change their behaviour in the absence of the dominant predators, then this is not coevolved niche partitioning but behavioural plasticity.	This sentence has been revised: "The diel temporal partitioning observed here in large coastal sharks is likely the result of a combination of constraints imposed by morphological or physiological limitations and behavioural plasticity."
15	278	As discussed in main comments, be clear whether you mean "physiological constraints", or "morphological constraints" instead of "evolutionary constraints"? Again, behaviour and its plasticity are evolved.	We have changed this to 'morphological constraints.'
16	283-284	As before I would suggest stating that this study specifically investigated diel temporal partitioning highlights the novelty of the findings, as there is a great deal of evidence for other forms of temporal partitioning in other taxa.	This has been corrected throughout.

Referee: 2

Comments to the Author(s)

Specific comments:

L 133 the name species of bull shark is *C. leucas*.

Thank you, this has been corrected.

L148 to 152. authors explain the time used by different sharks; however is very important to know what kind of prey they consume to know the shark species trophic habitat (bottom, pelagic, mesopelagic). Then authors must include background about the feeding habits of this shark species using published information. mainly from the study area. as an example *S. lewini* in the Gulf of California feed on benthic and coastal prey when they are juveniles (mainly fishes during the night), but when they are adults feed on squids in oceanic waters. Also *C. limbatus* predate mainly on fishes (sardines) in epipelagic coastal waters. Then that is the importance to give more information about the prey of these shark species in the study area to compare with the movements in the accelerometers.

Unfortunately, diet studies on the life stages/species examined here have not been conducted within the specific area of the study, however past work in nearby areas indicates all species are largely generalist teleost/elasmobranch predators. We have added the following text in lines 220-229 to touch upon the potential for diet overlap/resource partitioning:

“It is also notable that a degree of resource partitioning may also exist and may result directly from temporal partitioning, as the assemblage of available prey species likely varies throughout the diel cycle (e.g. Hagen and Able, 2008; Castillo-Rivera et al., 2010). Diet analyses have not been conducted on the sub-adult/adult populations of these species found within the specific area of the study, however, past work in nearby locations has determined that all species here are generalist teleost, elasmobranch, and cephalopod predators [55-59]. The only notable differences in diets between species in past studies are that marine reptiles, seabirds, and invertebrates (incl. molluscs and bivalves) may be important parts of the diet of tiger sharks [60], in addition to teleosts and elasmobranchs.”